# Exploring cytokine dynamics in tuberculosis: A comparative analysis of patients and controls with insights from three-week antituberculosis intervention

Michaela Krivošová[1]ʘ, Matúš Dohál[1]ʘ*, Simona Mäsiarová[2], Kristián Pršo[2], Eduard Gondáš[2], Radovan Murín[3], Soňa Fraňová[2], Igor Porvazník[4,5], Ivan Solovič[4,5], Juraj Mokrý[2]

1 Jessenius Faculty of Medicine in Martin, Biomedical Centre Martin, Comenius University Bratislava, Martin, Slovak Republic, 2 Jessenius Faculty of Medicine in Martin, Department of Pharmacology, Comenius University Bratislava, Martin, Slovak Republic, 3 Jessenius Faculty of Medicine in Martin, Department of Medical Biochemistry, Comenius University Bratislava, Martin, Slovakia, 4 National Institute for Tuberculosis, Lung Diseases and Thoracic Surgery, Vyšné Hágy, Slovak Republic, 5 Faculty of Health, Catholic University, Ružomberok, Slovak Republic

ʘ These authors contributed equally to this work.
* matus.dohal@uniba.sk

**Data Availability Statement:** All relevant data are within the paper and its Supporting Information.

## Abstract

Despite developing new diagnostics, drugs, and vaccines, treating tuberculosis (TB) remains challenging. Monitoring inflammatory markers can contribute to more precise diagnostics of TB, identifying its active and latent forms, or monitoring its treatment success. We assessed alterations in plasma levels of 48 cytokines in 20 patients (17 males) with active pulmonary TB compared to age-matched healthy controls (n = 18). Blood samples were collected from individuals hospitalised with TB prior to commencing antibiotic therapy, after the first week, and following the third week. The majority of patients received treatment with a combination of four first-line antituberculosis drugs: rifampicin, isoniazid, ethambutol, and pyrazinamide. Plasmatic cytokine levels from patients three times and controls were analyzed using a Bio-Plex Pro Human Cytokine Screening Panel. The results showed significantly higher levels of 31 cytokines (p<0.05) than healthy controls. Three-week therapy duration showed significantly decreased levels of nine cytokines: interferon alpha-2 (IFN-α2), interleukin (IL) 1 alpha (IL-1α), IL-1 receptor antagonist (IL-1ra), IL-6, IL-10, IL-12 p40, IL-17, leukemia inhibitory factor (LIF), and tumor necrosis factor alpha (TNF-α). Out of these, only levels of IL-1α and IL-6 remained significantly elevated compared to controls. Moreover, we have found a negative correlation of 18 cytokine levels with BMI of the patients but no correlation with age. Our results showed a clinical potential for monitoring the levels of specific inflammatory markers after a short treatment duration. The reduction in cytokine levels throughout the course of therapy could indicate treatment success but should be confirmed in studies with more individuals involved and a longer observation period.

**Funding:** This research was funded by grants of
the Slovak Research and Development Agency
(APVV-18-0084 - J.M.; APVV-22-0342 - J.M.),
grant of Scientific Grant Agency of the Ministry of
Education, Research, Development and Youth of
the Slovak Republic and Slovak Academy of
Sciences (VEGA-1/0093/22 - J.M.; VEGA-1/0042/
24 - E.G.). The funders had no role in study design,
data collection and analysis, decision to publish, or
preparation of the manuscript.

**Competing interests:** The authors have no
conflicts of interest to declare.

# 1. ntroduction

Tuberculosis (TB) remains a global health concern, with millions of new cases reported annually [1]. Accurate and timely diagnosis, coupled with effective treatment monitoring, is crucial for eradicating this infectious disease.

Traditional TB diagnostic methods often involve sputum microscopy and culture. Despite their clinical relevance, they have several limitations, such as time-consumption and low sensitivity, especially in extrapulmonary cases [2]. To address these limitations, several molecular genetic tests have been introduced into clinical practice in the last decade, such as Xpert MTB/RIF Ultra, Xpert MTB/XDR (Cepheid, California, USA), line probe assays, Deeplex® Myc-TB (based on targeted next-generation sequencing; Lille, France) as well as whole genome sequencing technologies [3–5]. Despite this, the high costs show a considerable obstacle to their application, primarily in developing countries with a high incidence of TB [6].

The emergence of blood biomarkers (cytokines and chemokines) provides a promising alternative, delivering a minimally invasive diagnostic approach that can surmount the limitations associated with traditional methods as they reflect the host's response to *Mycobacterium tuberculosis* (*Mtb*) infection [7,8]. Cytokines are secreted by various immune cells in response to infection, and their levels can be measured in different biological samples, such as blood or sputum [9,10]. In the context of TB, specific cytokines, including interferon-gamma (IFN-γ), tumour necrosis factor-alpha (TNF-α), interleukin (IL) 2 (IL-2), interferon gamma-induced protein-10 (IP-10) and IL-10, play a key role [11,12]. Elevated levels of pro-inflammatory cytokines, particularly IFN-γ and TNF-α, are associated with active TB, while IL-1 receptor antagonist (IL-1ra) and IL-10 are known for their immunosuppressive effects and have been linked to disease progression [13]. In addition to diagnosis, monitoring the effectiveness of TB treatment is equally important to ensure timely adjustments of the treatment regimen and prevent the emergence of drug resistance. Cytokine levels can be reliable biomarkers to assess host response to antituberculosis (anti-TB) treatment. A decrease in pro-inflammatory cytokine levels and a shift towards an anti-inflammatory profile during treatment indicate a positive response. Conversely, persistent or increasing cytokine levels may indicate treatment failure or the emergence of drug resistance, requiring reassessment of the therapeutic regimen [14–16].

The use of conventional diagnostic methods is limited especially in children, due to lack of sputum production and paucity or absence of microorganisms in respiratory secretions [17]. Recent results showed that the three-cytokine signature of TNF-α, IL-2, and IL-17A could serve as an accurate biomarker for the diagnosis of paediatric TB, which highlights the clinical potential of cytokine profiling [18].

In this study, we measured the plasmatic levels of 48 different cytokines in order to diagnose TB and predict the success of the ongoing treatment regimen.

## 2. Materials and methods

### 2.1. Study participants, blood sampling

In this prospective cohort study, 29 TB patients hospitalized between July 2021 and December 2022 at the National Institute of Tuberculosis, Lung Diseases and Thoracic Surgery in Vyšné Hágy, Slovakia, were included. The inclusion criteria were as following: age ≥ 18 years and diagnosis of active form of pulmonary TB confirmed by bacteriologic culture. Patients with latent or extrapulmonary forms and recurrent TB were excluded. Most of the patients underwent treatment with a combination of four first-line anti-TB drugs (rifampicin, isoniazid, ethambutol, pyrazinamide; HRZE regimen). All the patients were drug-susceptible. Their blood samples were collected as follows: at admission (hereinafter referred to as Day 1), in 7 days

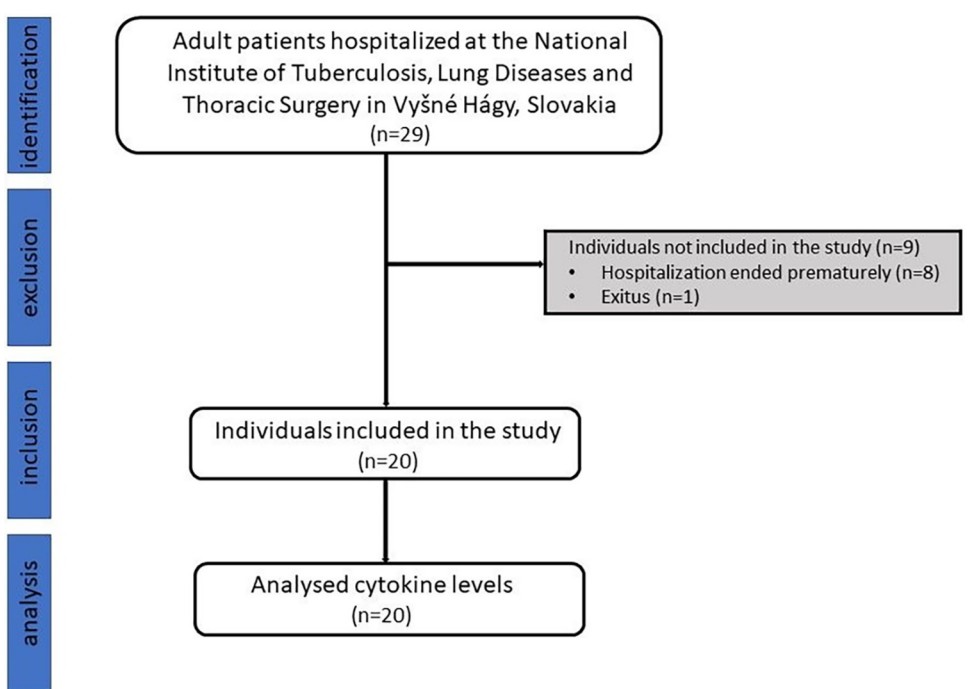

**Fig 1. STROBE flow diagram of the included patients.**

(hereinafter referred to as Week 1), and in 21 days (hereinafter referred to as Week 3). Blood was collected to EDTA tubes, centrifuged at 2500 × g for 15 min at room temperature and plasma was stored at −20°C until analysis. Patients whose hospitalization was terminated prematurely or who were discharged to ambulatory care (n = 8) were excluded from the study. One patient died before the completion of all sample collection. Out of the 20 remaining TB patients (age 46±12.4 years), 15 were administered first-line anti-TB regimen, while 5 patients received also second-line fluoroquinolones (levofloxacin or ofloxacin) due to drug intolerance. The inclusion procedure of the patients is showed in Fig 1. This study included also 18 age-matched healthy controls (9 males) with average age 46.9±18.2 years. Controls as well as patients in this study were Caucasians from Slovakia. Controls did not suffer from any infectious and inflammatory diseases, nor from any oncology diagnosis.

### 2.2. Ethical statement

Subjects were informed verbally and a written informed consent was obtained before their participation in the study. The study protocol, as well as the informed consent, was approved by the Ethical Committee of the Jessenius Faculty of Medicine in Martin, Comenius University Bratislava (EK65/2021). Prof. Mokrý, as the chairman of the ethics committee, was excluded from the voting about the decision on our project and, as a chairman, was substituted by Prof. Zibolen, as stated in the official document.

### 2.3. Multiplex Cytokine Assay

The analysis utilised 50 μl of plasma samples using Bio-Plex Pro Human Cytokine Screening Panel, 48-Plex kit (Bio-Rad, California, USA). Overview of the parameters analysed and their definition can be found in S1 Table. The subsequent analysis was conducted on the Bio-Plex 200 Systems (Bio-Rad, California, USA) in accordance with the manufacturer's instructions.

Briefly, the first step for determining inflammation markers was a sandwich enzyme immuno-assay (ELISA). The procedure involved the complexation of the analyte with magnetic beads and antibodies, utilizing the fluorescent reagent streptavidin-phycoerythrin within the micro-plate wells. Furthermore, immunocomplexes were identified using the flow system based on red radiation with a wavelength of 635 nm. Cytokines were monitored using green light with a wavelength of 532 nm.

## 2.4. Statistical analysis

Measured data were statistically analysed in GraphPad Prism 8.0.1 (GraphPad, San Diego, CA, USA). First, we identified and excluded outliers using the ROUT test (Q = 1%). We performed the Shapiro-Wilk test of normality, and the data were analysed accordingly. For normally distributed data, we used ordinary one-way ANOVA with Dunnett's multiple comparison test to compare levels in controls with those in patients at different times and RM one-way ANOVA with Tukey's multiple comparison test to analyse cytokine levels at different times. For non-parametric data, we used the Kruskal-Wallis test to compare controls and patients at different times and the Friedman test to observe dynamic changes in time. Both aforementioned analyses were followed by Dunn's multiple comparison test. Two-way ANOVA was used to compare the effect of different treatment regimens on cytokine levels. Spearman correlation was computed between continuous variables (age, BMI) and cytokine levels. The level of significance was set at $p<0.05$.

## 3. Results

### 3.1. Patients data

The primary reason for hospitalization was active TB, although some patients had also other concurrent comorbidities (see below). Nearly all of them had a history of smoking, and a majority acknowledged alcohol consumption. Certain comorbidities such as diabetes, hypertension, nephrological complications, and others were identified in some individuals (Tables 1 and S2). Considering these findings and the concurrent use of medications for associated conditions, a tailored treatment plan for TB was established and closely monitored.

### 3.2. Cytokine measurement

The levels of 5 cytokines (IL-2, IL-5, IL-15, nerve growth factor-beta (β-NGF), vascular endothelial growth factor (VEGF)) were undetectable or detectable only in a few samples, so they were excluded from further analysis.

Firstly, we have analysed the cytokine profile between patients with active pulmonary TB at admission (Day 1) and healthy controls. Out of 43 successfully determined cytokines in plasma, 31 were significantly increased in TB patients. Those were basic fibroblast growth factor (basic FGF), cutaneous T cell-attracting chemokine (CTACK), eotaxin, granulocyte colony stimulating factor (G-CSF), granulocyte macrophage-CSF (GM-CSF), growth-regulated onco-gene-alpha (GRO-α), hepatocyte growth factor (HGF), IFN-α2, IFN-γ, IL-1α, IL-1β, IL-1ra, IL-2Rα, IL-4, IL-6, IL-8, IL-12 p40, IL-13, IL-17, IL-18, interferon-gamma inducible protein-10 (IP-10), leukemia inhibitory factor (LIF), monocyte chemoattractant protein-1 (MCP-1), macrophage- CSF (M-CSF), macrophage migration inhibitory factor (MIF), monokine induced by gamma (MIG), macrophage inflammatory protein-1 alpha (MIP-1α), platelet-derived growth factor-BB (PDGF-BB), regulated on activation, normal T cell expressed and secreted factor (RANTES), stem cell growth factor-beta (SCGF-β), and TNF-α. There were no significantly decreased cytokine levels in TB patients. The data are shown in Fig 2.

**Table 1. Characteristics of the patients.**

| Total number of patients | | 20 |
|---|---|---|
| **Sex** | | |
| Female (%) | | 3 (15%) |
| Male (%) | | 17 (85%) |
| Age±SD (in years) | | 46±12.4 |
| **Body weight** | | |
| Obesity N (%) | | 1 (5%) |
| Slightly overweight N (%) | | 2 (10%) |
| Malnutrition N (%) | | 8 (40%) |
| BMI±SD | | 20±4.8 |
| **Abuses** | | |
| Smokers and ex-smokers N (%) | | 19 (95%) |
| Alcohol abusers N (%) | | 13 (65%) |
| **Comorbidities** | | |
| Disorders of central nervous system N (%) | | 4 (20%) |
| Cardiovascular disease N (%) | | 3 (15%) |
| Haematological problems N (%) | | 3 (15%) |
| Oncological diagnosis N (%) | | 2 (10%) |
| Kidney diseases N (%) | | 1 (5%) |
| Gastrointestinal diseases N (%) | | 2 (10%) |
| Other respiratory diseases N (%) | | 8 (40%) |
| Liver damage N (%) | | 5 (25%) |
| Metabolic disorders N (%) | | 6 (30%) |
| Infectious diseases | Covid N (%) | 2 (10%) |
| | Others N (%) | 7 (35%) |
| **Treatment outcome** | | |
| Cured successfully N (%) | | 19 (95%) |
| Exitus N (%) | | 1 (5%) |

N = number, SD–standard deviation, BMI = Body mass index.

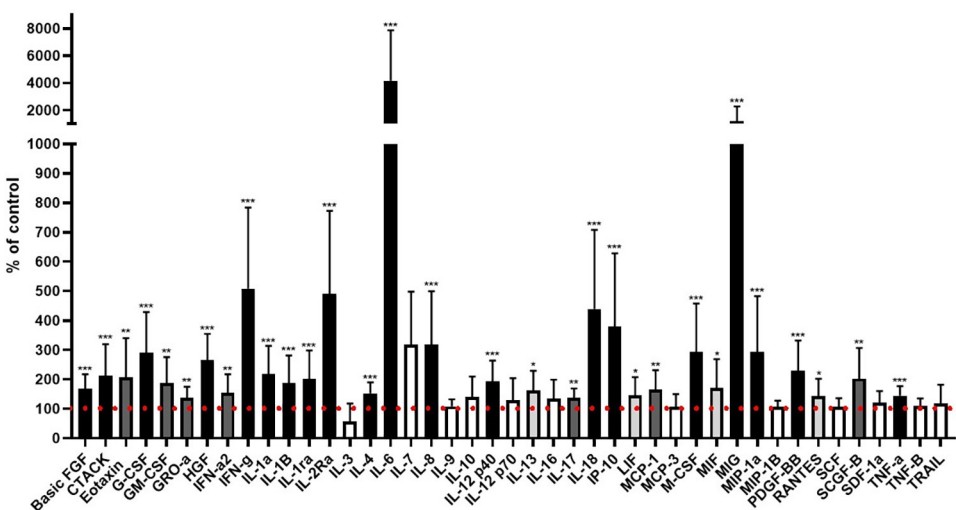

**Fig 2. Cytokine levels in patients with active pulmonary TB at admission (Day 1) compared to healthy controls.**
The data of TB patients are depicted as percentages of cytokine levels in controls. Boxes are further colour distinguished to represent the significance levels of the differences between patients and controls: Black *** (p<0.001), dark grey ** (p<0.01), light grey * (p<0.05), white–nonsignificant.

When analysing dynamic changes of cytokines during hospitalization and treatment with anti-TB drugs, levels of 9 cytokines have significantly decreased at Week 3 (IFN-α2, IL-1α, IL-1ra, IL-6, IL-10, IL-12 p40, IL-17, LIF, TNF-α). These results are shown in Fig 3.

During more detailed analysis of IL-10 levels, which was not elevated at baseline but showed significant decrease during the treatment, it was found that the classic HRZE regimen did not lead to significant change (p = 0.2090) while adding fluoroquinolones to the therapy led to significant decrease of this immunomodulatory cytokine (p = 0.0359).

We have not found any correlation between cytokine levels and age of the subjects in Day 1. However, when we correlated the levels with BMI of the patients, there was a significant negative correlation in levels of 18 cytokines with increasing BMI–basic FGF, G-CSF, IFN-α2, IFN-γ, IL-1α, IL-1ra, IL-2Ra, IL-8, IL-12 p70, IL-12 p40, IL-16, IL-18, IP-10, LIF, MCP-3, M-CSF, MIG, and MIP-1α. These are shown in Fig 4. As we did not have the weight and height updated during the therapy, we did not further correlate BMI with the treatment effect, which could be interesting in the future studies.

## 4. Discussion

Our study showed elevated levels of 31 cytokines in TB patients compared to healthy controls. Three-week anti-TB treatment led to decreased levels of 9 of them that could potentially reflect treatment efficacy.

The cytokine response in active pulmonary TB differs from that of other respiratory infections. In TB infection, it tends to be more prolonged and skewed towards Th1-type responses characterized by elevated IFN-γ levels [19]. However, some cytokines, such as TNF-α and IL-6, are not specific to TB and can be elevated in various infectious and inflammatory conditions. Additionally, the cytokine response in TB can vary depending on factors such as host immune status, disease severity, and presence of comorbidities [20,21].

The majority of studies analysed the long-term effect of anti-TB treatment [16,22–29]. Cytokines can contribute to both protective and pathological outcomes [14]. Initial elevation of cytokine levels is essential for early immune responses (IFN-α, IL-6), for survival following *Mtb* infection (TNF-α, IFN-γ, IL-1α, IL-1β, IL-12 p40, IL-17); while later on, they can potentiate tissue damage (TNF-α, IL-17), and induce other pathological pathways.

In this study, one-week interval was sufficient to observe a significant decrease in IL-17 levels and it was further significantly decreased also at Week 3. Fluctuations of IL-17 levels were also observed in TB patients in the study of Riou et al [28]; however, at a 26-week follow-up, the change was non-significant compared to the baseline levels. Furthermore, during primary TB, IL-17 with IFN-γ producing cells are usually induced and they promote the expression of other cytokines and chemokines and contribute to granuloma formation [30]. Therefore, it is necessary to achieve Th1 and Th17 balance to control bacterial growth and at the same time to limit the immunopathology caused by excessive IL-17 production.

Th1 and Th17 cells are the main effector CD4+ T cells in TB infection [19]. Major Th1 inducing cytokines include IFN-γ and IL-12, followed by IL-2, TNF-α, IL-12, IL-18, and IL-27. We found most of them significantly increased in TB patients compared to controls. Moreover, three weeks of anti-TB treatment caused significant decrease of TNF-α and IL-12 p40. Nie et al. observed that levels of IFN-γ and TNF-α could be used for monitoring the anti-TB treatment progress [31]. We have not detected a significant decrease in IFN-γ in our study, potentially caused by a short observation period.

IL-12 decreases TB bacterial burden [32] by maintaining IFN-γ production that limits long-term bacterial growth [33]. Their levels are mediated by IL-10 [24]. IL-12 p40 deficient mice were found to be *Mtb* infection susceptible [34]. Our study showed a notable reduction

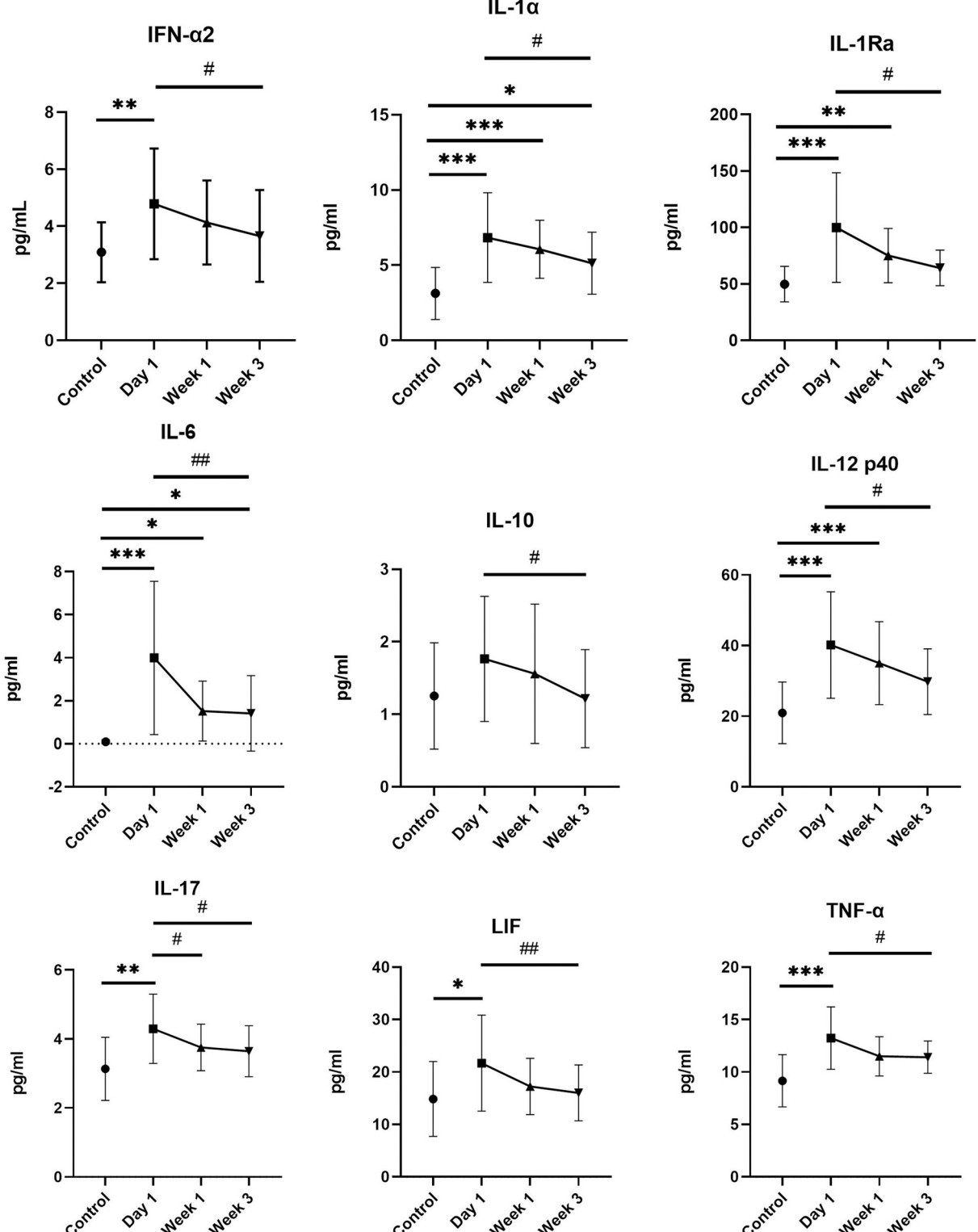

**Fig 3. Dynamic changes (Day 1, Week 1, Week 3) in cytokine levels in patients during anti-TB treatment in hospital.** Only cytokines with significantly changed levels during hospitalization are showed. * indicates differences between controls and patients in different times and # shows differences in patients in time.

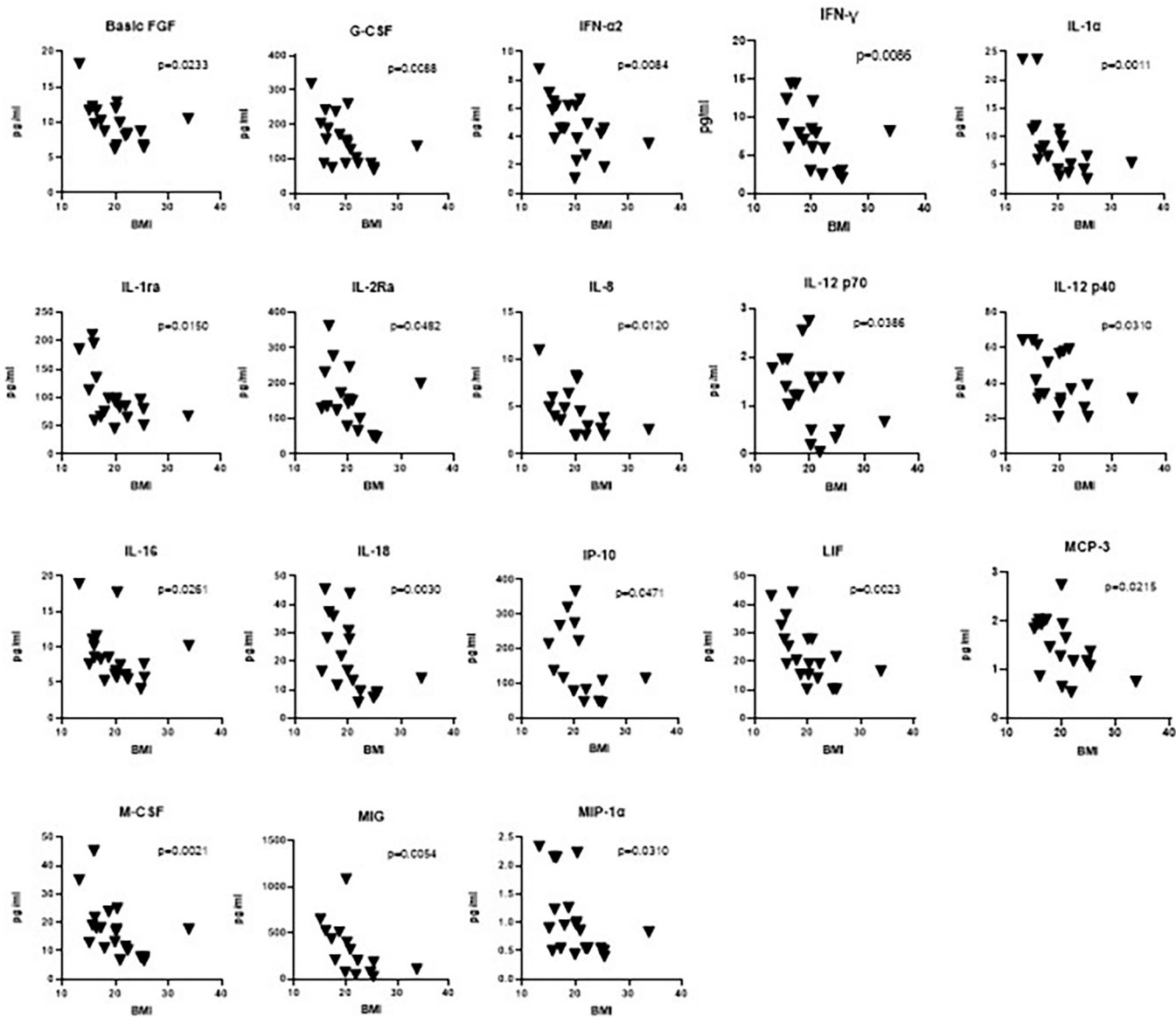

**Fig 4. Significant correlation between cytokine levels and BMI of patients in Day 1.**

in levels of IL-12 p40 in three weeks. A significant reduction of this cytokine was detected also long-term, after 6-month anti-TB therapy in the study of Chowdhury et al [26].

Furthermore, individuals with TB exhibited elevated levels of circulating IL-1ra and IL-1α, which have been proposed as potential biomarkers for diagnosing the active disease or evaluating the response to treatment [35]. IL-1α and IL-1ra are two related but functionally distinct proteins involved in the regulation of inflammatory processes [36]. The IL-1α /IL-1ra balance is considered as critical in TB infection. IL-1ra inhibits the action of proinflammatory IL-1α by competitively blocking IL-1R1 receptors [37]. IL-1ra was previously found to be decreased over the course of anti-TB treatment in active as well as latent TB [38,39]. In our study, levels of both IL-1α and IL-1ra decreased after three-week therapy during hospitalisation.

IL-6 was identified as a potential marker of early TB treatment response, thus holds promise for treatment monitoring [16,26]. The authors measured IL-6 levels at a 6-week follow-up and

it helped them distinguish between the patients with positive and negative prognosis [16]. In our study, we have observed a significant decrease after three-week anti-TB treatment, which is consistent with both previously mentioned studies. However, it is premature to conclude whether IL-6 can effectively distinguish treatment outcomes already at Week 3, as all but one patients in our study improved and recovered after anti-TB therapy.

IL-10 is an immunomodulatory cytokine and inhibits Th1 cell differentiation. Its higher levels in TB have been associated with the disease progression and poor disease outcome [40]. In this study, patients did not have significantly higher levels of this anti-inflammatory cytokine; however, we have observed a significant decrease after three-week treatment with anti-TB drugs. We further correlated the levels and its dynamics in patients who were administered classical HRZE therapy and who had to take second-line anti-TB–fluoroquinolones. Interestingly, only in the group of patients taking fluoroquinolones (ofloxacine, levofloxacine), the levels of IL-10 significantly decreased. This finding supports previously observed results showing that fluoroquinolones have specific immunomodulatory effects [41,42].

Three-week anti-TB therapy led also to significantly decreased levels of IFN-α2 and LIF that were found increased at baseline compared to healthy controls. In previous study, LIF was similarly significantly increased in patients with both active and latent TB [43]. It is considered as a key factor preventing alveolar injury. When removed in animal models of pneumonia, it has been linked to acute respiratory distress syndrome (ARDS) coupled with sepsis. Moreover, LIF was shown to protect against cytokine storm [44]. IFN-α2 is a type I IFN cytokine, essential for the initial production of protective IL-12 and TNF-α. However, its high levels inhibit their production, induce IL-10 secretion, which leads to suppression of IFN-γ–dependent host-protective immune responses [45], thus promotes both bacterial expansion and disease pathogenesis. Their decrease after three weeks of therapy might predict successful anti-TB therapy.

Similarly to the results of Chowdhury et al. we observed a negative correlation with BMI but none correlation with age during a correlation analysis of these parameters with cytokine levels [26]. Specifically, 18 cytokines were significantly increased with decreasing BMI at baseline. Almost half of the patients in this study were cachectic, reflecting negative energy balance in TB. It is a consequence of chronic systemic inflammatory condition caused by this specific infection [46].

This study has several limitations. Short observation period and small number of patients are the major ones. Levels of cytokines can fluctuate during the anti-TB treatment [28]; therefore, an initial reduction in cytokines does not necessarily indicate ultimate improvement following the therapy. Longer follow-up would provide a more comprehensive understanding of how cytokine levels evolve throughout the treatment process. However, even studies with longer observation period failed to observe significant changes, which could reflect the compartmentalized production of cytokines and continuous decline at the site of infection in the lungs [28]. Moreover, we are aware that such small sample size cannot adequately represent the diverse population of TB patients considering also their various comorbidities. Another minor limitation of this study can be gender disproportion between patients and healthy controls.

## 5. Conclusion

In conclusion, our study sheds light on cytokine dynamics during the initial three weeks of anti-TB treatment and identifies elevated cytokines in active pulmonary TB compared to healthy controls. We observed significant reduction in levels of 9 cytokines, including TNF-α, IL-12 p40, IFN-α2, and LIF, after three weeks of therapy, highlighting their potential of early treatment monitoring. Additionally, our correlation analysis suggests a potential association

between cytokine levels and BMI, emphasizing the importance of considering host factors in TB pathogenesis and treatment response. Such analysis could be further performed in paediatric patients to compare the infection dynamics and establish the cytokine profiling for TB diagnosis and treatment response in younger population.

## Supporting information

**S1 Table. List of included parameters and their definition.**
(DOCX)

**S2 Table. Summary of comorbidities of included patients.**
(DOCX)

**S1 Dataset.**
(XLSX)

## Author Contributions

**Conceptualization:** Michaela Krivošová, Matúš Dohál, Eduard Gondáš, Radovan Murín.

**Data curation:** Igor Porvazník.

**Formal analysis:** Kristián Pršo, Ivan Solovič.

**Supervision:** Soňa Fraňová, Ivan Solovič, Juraj Mokrý.

**Writing – original draft:** Michaela Krivošová, Matúš Dohál, Simona Mäsiarová.

**Writing – review & editing:** Soňa Fraňová, Juraj Mokrý.

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
