## [Editor Report · Decision Letter 0]

10 Apr 2024

PONE-D-24-08188Exploring Cytokine Dynamics in Tuberculosis: A Comparative Analysis of Patients and Controls with Insights from Three-Week Antituberculosis InterventionPLOS ONE

Dear Dr. Dohál,

Thank you for submitting your manuscript to PLOS ONE. After careful consideration, we feel that it has merit but does not fully meet PLOS ONE’s publication criteria as it currently stands. Therefore, we invite you to submit a revised version of the manuscript that addresses the points raised during the review process.

This is a good study. 

There is novelty in this study. 

I give some comments for improvement before proceed to the reviewers. 

- Please improve the quality of the figure. 

- Please give detail the design of your study.

- Please begin the discussion with the main findings of your study. 

- Please elaborate more the limitations of your study. 

- The discussion section is too long.

- The conclusion should be concise and clear. 

We look forward to receiving your revised manuscript.

Kind regards,

Rizaldy Taslim Pinzon

Academic Editor

PLOS ONE

“This research was funded by grants of the Slovak Research and Development Agency (APVV-18-0084; APVV-22-0342), grant of Scientific Grant Agency of the Ministry of Education, Research, Development and Youth of the Slovak Republic and Slovak Academy of Sciences (VEGA-1/0093/22).”

3. In the online submission form, you indicated that [All relevant data are within the manuscript and its Supporting Information files may be shared upon request].

Additional Editor Comments:

This is a good study.

There is novelty in this study.

I give some comments for improvement before proceed to the reviewers.

- Please improve the quality of the figure.

- Please give detail the design of your study.

- Please begin the discussion with the main findings of your study.

- Please elaborate more the limitations of your study.

- The discussion section is too long.

- The conclusion should be concise and clear.

---

## [Author Response · Author response to Decision Letter 0]

15 Apr 2024

RESPONSE TO REVIEWERS

Thank you very much for the initial review of our study and for your suggestions for further improvement. Here, we react to all your comments and the relevant revisions have been made also in the manuscript. All authors approved the revisions and the latest version of the manuscript.

This is a good study. 

There is novelty in this study. 

I give some comments for improvement before proceed to the reviewers. 

- Please improve the quality of the figure. 

We have improved the quality of the figures 3 and 4 and uploaded the newer versions. Thank you for this suggestion.

- Please give detail the design of your study.

We have added the design of this study in the Methods section as well as inclusion and exclusion criteria. The following sentences have been edited:

line 72 - In this prospective cohort study …

line 74-76 - The inclusion criteria were as following: age ≥ 18 years and diagnosis of active form of pulmonary TB confirmed by bacteriologic culture. Patients with latent or extrapulmonary forms and recurrent TB were excluded.

- Please begin the discussion with the main findings of your study. 

Thank you for this comment. We began the discussion with the following sentences: 

Our study showed elevated levels of 31 cytokines in TB patients compared to healthy controls. Three-week anti-TB treatment led to decreased levels of 9 of them that could potentially reflect treatment efficacy. (line 163-164)

- Please elaborate more the limitations of your study. 

We have edited the limitations of the study as follows:

line 259 - This study has several limitations.

line 262-263 - Longer follow-up would provide a more comprehensive understanding of how cytokine levels evolve throughout the treatment process.

line 265-267 - Moreover, we are aware that such small sample size cannot adequately represent the diverse population of TB patients considering also their various comorbidities.

- The discussion section is too long.

The discussion section was shortened. We updated the references accordingly.

- The conclusion should be concise and clear. 

We reworded the conclusion section making it more concise and highlighting the main findings:

In conclusion, our study sheds light on cytokine dynamics during the initial three weeks of anti-TB treatment and identifies elevated cytokines in active pulmonary TB compared to healthy controls. We observed significant reduction in levels of 9 cytokines, including TNF-α, IL-12 p40, IFN-α2, and LIF, after three weeks of therapy, highlighting their potential of early treatment monitoring. Additionally, our correlation analysis suggests a potential association between cytokine levels and BMI, emphasizing the importance of considering host factors in TB pathogenesis and treatment response. Such analysis could be further performed in paediatric patients to compare the infection dynamics and establish the cytokine profiling for TB diagnosis and treatment response in younger population. (line 270-277)

---

## [Decision Letter · Decision Letter 1]

14 May 2024

PONE-D-24-08188R1Exploring Cytokine Dynamics in Tuberculosis: A Comparative Analysis of Patients and Controls with Insights from Three-Week Antituberculosis InterventionPLOS ONE

Dear Dr. Dohál,

Thank you for submitting your manuscript to PLOS ONE. After careful consideration, we feel that it has merit but does not fully meet PLOS ONE’s publication criteria as it currently stands. Therefore, we invite you to submit a revised version of the manuscript that addresses the points raised during the review process.

Please elaborate more this comments :The study's small sample size and short observation period significantly undermine the strength and applicability of the findings. With only 20 TB patients studied over three weeks, it is difficult to generalize these results to the broader TB patient population. The conclusions drawn from such a limited dataset risk being premature, particularly concerning the efficacy and durability of treatment responses.

We look forward to receiving your revised manuscript.

Kind regards,

Rizaldy Taslim Pinzon

Academic Editor

PLOS ONE

Journal Requirements:

Additional Editor Comments:

Thank you authors for the prompt reply.

Please elaborate more this comments.

The study's small sample size and short observation period significantly undermine the strength and applicability of the findings. With only 20 TB patients studied over three weeks, it is difficult to generalize these results to the broader TB patient population. The conclusions drawn from such a limited dataset risk being premature, particularly concerning the efficacy and durability of treatment responses.

Reviewers' comments:

Reviewer's Responses to Questions

**Comments to the Author**

1. If the authors have adequately addressed your comments raised in a previous round of review and you feel that this manuscript is now acceptable for publication, you may indicate that here to bypass the “Comments to the Author” section, enter your conflict of interest statement in the “Confidential to Editor” section, and submit your "Accept" recommendation.

Reviewer #1: All comments have been addressed

Reviewer #2: All comments have been addressed

2. Is the manuscript technically sound, and do the data support the conclusions?

Reviewer #1: Yes

Reviewer #2: Yes

3. Has the statistical analysis been performed appropriately and rigorously? 

Reviewer #1: Yes

Reviewer #2: Yes

4. Have the authors made all data underlying the findings in their manuscript fully available?

Reviewer #1: Yes

Reviewer #2: Yes

5. Is the manuscript presented in an intelligible fashion and written in standard English?

Reviewer #1: Yes

Reviewer #2: (No Response)

6. Review Comments to the Author

Reviewer #1: The study addresses a critical gap in TB treatment by exploring cytokine levels as potential biomarkers for monitoring the disease's progression and response to treatment. The results are quite interesting; however, the paper has several notable deficiencies that detract from its overall impact:

1 The study's small sample size and short observation period significantly undermine the strength and applicability of the findings. With only 20 TB patients studied over three weeks, it is difficult to generalize these results to the broader TB patient population. The conclusions drawn from such a limited dataset risk being premature, particularly concerning the efficacy and durability of treatment responses.

2. While the study includes a control group, further details about its selection and characteristics could help strengthen the comparative analysis. It is crucial to ensure that the control group closely matches the patient's demographic status. I couldn't find such information in the text.

3. The study does not consider the complex interactions between different cytokines or external factors that might influence cytokine levels, such as concurrent infections or underlying chronic conditions. This oversimplification can lead to misleading conclusions about the role and significance of specific cytokines in TB treatment response.

4. The study finds a correlation between cytokine levels and BMI, suggesting that nutritional and metabolic factors affect the immune response to TB. Expanding this research to include more comprehensive metabolic profiling could uncover additional insights into how these factors influence treatment outcomes

5. While the study focuses on cytokines as markers of treatment efficacy, exploring their role in predicting treatment failure or complications could also be valuable. Additionally, examining cytokine profiles in conjunction with other biomarkers, such as genetic or molecular markers, might provide a more holistic view of the disease and its management

Reviewer #2: The study has novelty and can be published. It will be a useful addition for further studies and research. Though clinical parameters are of paramount importance, the inflammatory markers may help aid the prognosis at an early stage.

7. PLOS authors have the option to publish the peer review history of their article (what does this mean?). If published, this will include your full peer review and any attached files.

Reviewer #1: No

Reviewer #2: **Yes: **Dr Gyanshankar Mishra

---

## [Author Response · Author response to Decision Letter 1]

16 May 2024

1. The study's small sample size and short observation period significantly undermine the strength and applicability of the findings. With only 20 TB patients studied over three weeks, it is difficult to generalize these results to the broader TB patient population. The conclusions drawn from such a limited dataset risk being premature, particularly concerning the efficacy and durability of treatment responses:

Dear Editor and Reviewer, thank you very much for this helpful comment. We are aware of the small cohort of patients, so we have included this information in the study limitations. However, this is due to several factors: 

1. Slovakia is among the countries with a low incidence of tuberculosis (an average of 130 cases pulmonary tuberculosis per year), of which approximately 40% of cases are diagnosed in the pediatric population, which was not included in the study. 

2. The study was conducted in collaboration with the facility with the highest number of hospitalized patients in Slovakia: National Institute for Tuberculosis, Lung Diseases and Thoracic Surgery, Vyšné Hágy, Slovak Republic. However, some patients are also hospitalized in regional hospitals, which has contributed to a lower cohort of patients, as we do not actively cooperate with these facilities (many of them are private).

3. Patients with drug-resistant tuberculosis were excluded due to prolonged treatment regimens and the expected slower reflection of the effectiveness of therapy on the cytokine profile.

4. Patients with extrapulmonary TB and HIV were also excluded.

Regarding the observation period, following the improvement of the condition (in case of drug-susceptible tuberculosis), patients are often discharged for outpatient treatment three weeks after the initiation of therapy. To ensure uniform sampling times and to collect samples from as many patients as possible, we opted to sample on the day treatment commenced and after 7 and 21 days. 

While the study includes a control group, further details about its selection and characteristics could help strengthen the comparative analysis. It is crucial to ensure that the control group closely matches the patient's demographic status. I couldn't find such information in the text.

Thank you for this comment regarding the control group. The individuals included in this group were carefully selected to be age-matched with the patients and the equal number of females and males was chosen to make this group representative. Controls as well as patients in this study were Caucasians from Slovakia. Controls did not suffer from any infectious and inflammatory diseases, nor from any oncology diagnosis. 

The study does not consider the complex interactions between different cytokines or external factors that might influence cytokine levels, such as concurrent infections or underlying chronic conditions. This oversimplification can lead to misleading conclusions about the role and significance of specific cytokines in TB treatment response.

Considering the potential interactions between cytokines and external factors, in such a small group with variable comorbidities we did not further divide them into subgroups according to their underlying conditions as the outcomes would not be sufficiently representative. We prepared a table summarizing all comorbidities for each patient (see Supplementary Table 2.). We considered only the effects of various treatment regimens on the levels of cytokines, which has been mentioned in the study. Also, due to the pleiotropic nature of cytokines and their complex interactions, correlation analysis would likely produce numerous results without yielding any significant conclusions.

The study finds a correlation between cytokine levels and BMI, suggesting that nutritional and metabolic factors affect the immune response to TB. Expanding this research to include more comprehensive metabolic profiling could uncover additional insights into how these factors influence treatment outcomes

Thank you for this comment. We agree that inclusion of comprehensive metabolic profiling could bring new important insights into the pathophysiology and treatment impact of pulmonary TB. In the future, we want to expand the patient cohort to include pediatric patients and metabolomic analyses.

While the study focuses on cytokines as markers of treatment efficacy, exploring their role in predicting treatment failure or complications could also be valuable. Additionally, examining cytokine profiles in conjunction with other biomarkers, such as genetic or molecular markers, might provide a more holistic view of the disease and its management.

This is actually our plan for future studies. We have started with the analysis of genetic (SNPs) and epigenetic markers (miRNAs) related to the diagnosis of TB and we are also expanding our sample size. 

Reviewer #2: The study has novelty and can be published. It will be a useful addition for further studies and research. Though clinical parameters are of paramount importance, the inflammatory markers may help aid the prognosis at an early stage.

We would like to thank Reviewer 2 for the time dedicated to the review of our manuscript and for the positive comments.

---

## [Decision Letter · Decision Letter 2]

27 May 2024

Exploring Cytokine Dynamics in Tuberculosis: A Comparative Analysis of Patients and Controls with Insights from Three-Week Antituberculosis Intervention

PONE-D-24-08188R2

Dear Dr. Dohal

We’re pleased to inform you that your manuscript has been judged scientifically suitable for publication and will be formally accepted for publication once it meets all outstanding technical requirements.

Kind regards,

Rizaldy Taslim Pinzon

Academic Editor

PLOS ONE

Additional Editor Comments (optional):

Thank you authors for the appropriate responds.

Reviewers' comments:

Reviewer's Responses to Questions

**Comments to the Author**

1. If the authors have adequately addressed your comments raised in a previous round of review and you feel that this manuscript is now acceptable for publication, you may indicate that here to bypass the “Comments to the Author” section, enter your conflict of interest statement in the “Confidential to Editor” section, and submit your "Accept" recommendation.

Reviewer #1: (No Response)

Reviewer #2: All comments have been addressed

2. Is the manuscript technically sound, and do the data support the conclusions?

Reviewer #1: Yes

Reviewer #2: Yes

3. Has the statistical analysis been performed appropriately and rigorously? 

Reviewer #1: Yes

Reviewer #2: Yes

4. Have the authors made all data underlying the findings in their manuscript fully available?

Reviewer #1: Yes

Reviewer #2: Yes

5. Is the manuscript presented in an intelligible fashion and written in standard English?

Reviewer #1: Yes

Reviewer #2: Yes

6. Review Comments to the Author

Reviewer #1: The entirety of the responses and modifications contribute to my positive consideration of the manuscript for publication.

Reviewer #2: The authors have addressed the review comments noted previously. The article can be published. It will be a useful addition to literature. Lab markers can useful to gauge treatment response along with clinical evaluation.

7. PLOS authors have the option to publish the peer review history of their article (what does this mean?). If published, this will include your full peer review and any attached files.

Reviewer #1: No

Reviewer #2: No

---

## [Editor Report · Acceptance letter]

20 Jul 2024

PONE-D-24-08188R2 

PLOS ONE

Dear Dr. Dohál, 

I'm pleased to inform you that your manuscript has been deemed suitable for publication in PLOS ONE. Congratulations! Your manuscript is now being handed over to our production team.

Kind regards, 

on behalf of

Dr. Rizaldy Taslim Pinzon 

Academic Editor

PLOS ONE